# An Enhanced Photogrammetric Approach for the Underwater Surveying of the Posidonia Meadow Structure in the Spiaggia Nera Area of Maratea

**DOI:** 10.3390/jimaging9060113

**Published:** 2023-05-31

**Authors:** Francesca Russo, Silvio Del Pizzo, Fabiana Di Ciaccio, Salvatore Troisi

**Affiliations:** 1International PhD Programme, UNESCO Chair “Environment, Resources and Sustainable Development”, Department of Science and Technology, Parthenope University of Naples, 80143 Naples, Italy; francesca.russo008@studenti.uniparthenope.it (F.R.); fabiana.diciaccio@studenti.uniparthenope.it (F.D.C.); salvatore.troisi@uniparthenope.it (S.T.); 2Prisma S.r.l, 80065 Sant’Agnello, NA, Italy

**Keywords:** 3D modeling, environment, image enhancements, monitoring operations, *Posidonia oceanica*, underwater photogrammetry

## Abstract

The *Posidonia oceanica* meadows represent a fundamental biological indicator for the assessment of the marine ecosystem’s state of health. They also play an essential role in the conservation of coastal morphology. The composition, extent, and structure of the meadows are conditioned by the biological characteristics of the plant itself and by the environmental setting, considering the type and nature of the substrate, the geomorphology of the seabed, the hydrodynamics, the depth, the light availability, the sedimentation speed, etc. In this work, we present a methodology for the effective monitoring and mapping of the *Posidonia oceanica* meadows by means of underwater photogrammetry. To reduce the effect of environmental factors on the underwater images (e.g., the bluish or greenish effects), the workflow is enhanced through the application of two different algorithms. The 3D point cloud obtained using the restored images allowed for a better categorization of a wider area than the one made using the original image elaboration. Therefore, this work aims at presenting a photogrammetric approach for the rapid and reliable characterization of the seabed, with particular reference to the Posidonia coverage.

## 1. Introduction

*Posidonia oceanica* has a fundamental role for marine ecosystems: it is considered a natural carbon sink thanks to its capacity to store large amounts of carbon in its sediments over long periods [1]. Moreover, the “matte” of Posidonia constitutes a complex habitat in which various organisms can find nurseries and protection [2]. The accurate monitoring of the Posidonia extension and growth over the years assumes particular importance for the assessment of the biodiversity’s health status and its preservation.

In the past decades, the joint use of acoustic and optical data has proved to be a reliable tool to obtain high-resolution thematic maps, which are further used for the preliminary characterization of marine phanerogam habitats [3]. Moreover, direct measurement methods carried out by an underwater technical operator and/or with video inspections and images have always represented the most effective method for validating the data acquired with indirect instruments. 

The current direct methodologies adopted for the survey of Posidonia are defined by the ISPRA guidelines [4], which are used for the characterization of the habitat condition as required by the Marine Strategy Monitoring Programs (Art. 11, Legislative Decree 190/2010), implementation of the European legislation for species, and marine habitats monitoring of the Directives 92/43/CE “Habitats” and 2009/147/CE “Birds”. 

These operations generally involve divers in the collection of different kinds of data that will be used for laboratory analysis; the latter aim at defining the state of the health of the Posidonia meadows, the seabed coverage, the type of sediments, etc. [5].

The characterization of the seabed coverage, performed in this work, was made following the ISPRA protocol [4]. In the context of the seabed investigation, ISPRA requires the definition of a 15 m deep station and the identification of the lower limits of the meadows; the divers will then identify the working site within a circular area with a 5 m radius (to be measured with respect to a chosen fixed point). In each station, the estimations are made for both the type of substrate and the percentage of dead matte, of *C. racemosa* and of *C. nodosa* (see Table 1); moreover, the density measurements and the visual data for morphometric, lepidochronological, and biomass analysis are provided [4]. 

Table 1 shows the different parameters for visual estimation at sea. This is usually carried out simultaneously by two operators, and the average between their estimates then provides the coverage value [6].

**Table 1 jimaging-09-00113-t001:** Visual estimation parameters at sea (adapted from [7]).

Parameter	Measurement Unit
Meadow continuity	1 = continuous, 2 = discontinuous
% dead matte coverage	%
% alive PO coverage	%
% Caluerpa racemosa	%
% Cymodocea nodosa	%
Substrate	1 = rocks, 2 = sand, 3 = matte
Disturbance factors	1 = yes; 2 = none
Meadow composition	1 = pure; 2 = mixed
Non-native algae	1= Caluerpa racemosa; 2= Caluerpa taxifolia; 3 = both

However, visual sensors working underwater are affected by various environmental factors, which alter their perception and produce color-distorted images. In fact, they often suffer from severe quality degradation due to light absorption and scattering in water, resulting in low contrast and very fuzzy details with a hue that tends to be close to green and blue. 

This study proposes a photogrammetric approach for the investigation of the seabed morphology to support the normal operations required by the ISPRA guidelines, which is further improved by the use of two different single-image enhancement algorithms. More specifically, underwater photogrammetry is used here to produce accurate visual estimates of the *Posidonia oceanica* meadow coverage, in line with the standard current methodologies [6]. In fact, the visual assessment made by the diver on a limited surface can be integrated with a larger area analysis on the point cloud obtained by the photogrammetric elaboration of video frames; the latter can be acquired by the diver themselves or by other means, such as cameras mounted on remotely operated vehicles (ROVs) and autonomous underwater vehicles (AUVs).

The analysis of the point cloud also allows to estimate the height and profile of a matte with an accuracy that highly satisfies the ISPRA guidelines requirements; these measures are performed using an open-source software for point cloud management: CloudCompare. Following the preliminary estimations of the Posidonia “matte” presented by Russo et al. [8], in this work (i) the image blurring and light absorption (IBLA) algorithm [9] and (ii) an enhanced convolutional neural network (CNN)-based deep learning (DL) method [7] were separately used on the frames of each transect. These algorithms allowed a more accurate estimate of the background light and of the underwater scene depth, which helped in the restoration of the image’s true colors and provided for a more accurate 3D model reconstruction, from which more reliable considerations have been further made on the seabed coverage characterization.

The results obtained from this combined approach acquire particular importance in view of future bio-ecological operations involving reforestation interventions. In fact, studies carried out on transplanted areas show that Posidonia can take root with greater possibility in areas with the presence of dead matte [10]. In this regard, the combined use of our photogrammetric approach with conventional survey techniques could provide a reliable and useful means to improve the knowledge of the matte extension and further guide the reforestation interventions while monitoring their development.

This paper is organized as follows: The next section presents a brief overview of underwater monitoring applications and underwater photogrammetry, including the issues related to this particular environment’s operativity, with particular reference to those related to image acquisition and elaboration. Section 2 gives some technical information on the area and the instrumentation employed for the surveys, the methodology, and the description of the algorithms chosen for the image restoration phase. Section 3 presents the details of the photogrammetric elaboration and the obtained results, which are further compared with those obtained after the image enhancement step; some considerations finally conclude the paper.

### State of the Art

As previously introduced, the monitoring of the Posidonia meadows is currently performed by divers who methodically acquire direct measurements of the limits, dimensions, and erosion of the matte [11,12]. This process has great reliability but is limited by the operator’s dive time, the high costs, and the campaign completion time. For this reason, many other supportive applications are tested to make these monitoring procedures cheaper while maintaining their reliability and accuracy.

Side-scan sonar (SSS) and multi-beam echosounder (MBES) are considered the most common acoustic methods for seafloor mapping and analysis; AUV-based methodologies can also be used to detect and map *Posidonia oceanica* with good results, but with some limitations related to the higher costs of deployment [13,14].

Photogrammetric techniques allow to reconstruct a three-dimensional (3D) object from its images (2D) acquired from different poses [15]. In this context, the structure-from-motion (SfM) methodology has been developed for terrestrial settings and, by virtue of its very satisfying results, has also been employed as an important tool for creating three-dimensional models of underwater morphology and habitats [16,17]. Its application in monitoring and investigation shows excellent results also in the direct study of protected biocenoses, such as *Posidonia oceanica*, thanks to the easy interpretability of the investigated seabed data [18,19,20].

In fact, underwater photogrammetry allows the virtual reconstruction of 3D models and/or the obtaining of orthorectified photo assemblages (orthomosaics) of the sea floor through the processing of a large number of images of the area of interest. These frames can be acquired by a diver [21], by an AUV [22,23], an ROV [24], or by a towed camera [25] in a non-invasive approach. Important elements of this latter system include the integration of multiple cameras and a positioning system based on the global navigation satellite system (GNSS), similar to those used in underwater positioning systems (i.e., ultra-short baseline, long baseline, etc.) [26]. In this way, the experts are provided with more and easier-to-read data, especially when the acoustic survey results are difficult to interpret. 

The first experiments in the field of underwater photogrammetry started in the 1960s for the detailed study of archaeological finds on the seabed. Since then, interest in this technique in the field has been growing with the arrival of digital technology, thanks also to the implementation of geographic information systems (GIS) and the creation of digital terrain models (DTM). The book “3D Recording and Interpretation for Maritime Archaeology” [27] reports several contributions to the topic and numerous reviews in which techniques for underwater measurements are explored. Among them, some examples can be found in the creation of networks of control points for an accurate positioning system or in the creation of 2D models and 3D reconstructions of wrecks, with a comparison among different approaches [28,29,30,31].

Regardless of the scope of the photogrammetric survey, it is important to carefully consider the effect of several environmental factors that can affect the use of the chosen technique, such as the turbidity of the water, which influences the chromatic effect of the images [32]. In-depth studies on the detection and monitoring of environmental changes have been conducted in the marine ecology field, with the scope of studying environments such as coral reefs [31] or benthic environments [14] to further monitor their positive or negative evolution over the years. Their approach usually involves the use of multiple cameras combined with navigation and positioning systems [14] or the use of algorithms for image enhancement [16].

Underwater photogrammetry is often associated with shallow water monitoring, where positioning and measurement of targets with established positioning systems are still possible [14]. The use of dynamic positioning systems associated with ROVs can certainly be a good compromise, also in terms of the precision of positioning data, as long as these systems can provide continuous and valid data for the entire duration of the navigation. A multidisciplinary approach can also be adopted when performing acoustic surveys to be combined with satellite images [33]. A fundamental issue concerns the association of high-resolution images acquired with the cameras and the related positioning data; in this case, the problem is usually solved by using highly specialized and expensive tools, which are unfortunately not always within the reach of researchers.

Image enhancement techniques are often applied to reduce the image’s greenish-blue component and to obtain the structural and morphological aspects of *P. oceanica* meadow; this is usually performed by determining the surfaces covered by the various biocenoses and the dimensions of the investigated area on a scaled image [34]. These thematic maps can be used for time-lapse comparisons that quantify the changes in the seabed cover, such as those caused by anthropogenic impacts (for example, trawling and boat anchoring), and also to evaluate the blue carbon sinks, which further allow to define useful specific conservation strategies for future seagrass habitats [16].

#### Underwater Image Restoration

As previously mentioned, the quality of underwater images is severely affected by the physical and chemical characteristics of the environment. They usually show a color cast due to the different attenuation ratios of red, green, and blue lights in water. In particular, red light tends to be strongly absorbed when it enters the water, an effect that rapidly decreases in intensity, while green and blue light (having shorter wavelengths) can penetrate deeper below sea level, resulting in a greenish and blueish effect. Moreover, the majority of light energy is absorbed by the suspended particles and further diffracted before reaching the camera, which leads to images with low contrast, blur, and haze [35]. This is usually overcome using artificial light sources, but those are still affected by absorption and scattering, other than by introducing shadowing and non-uniform illumination, which results in a bright, centered spot that decreases towards the image edges [36]. Other issues may occur due to the influence of water turbidity. The purpose of the image enhancement and restoration algorithms is to correct the color, blurring, and background scattering to improve the image quality and extract valuable information. The visibility of underwater images can be improved using hardware [37] and software solutions [38]. In the first case, hardware platforms and cameras tend to be power-consuming and expensive, other than being poorly adaptable to underwater scenarios; for this reason, algorithmic approaches have been developed. Some works can be cited, for example, Kaeli et al. [39], which focused on algorithms for underwater image color correction, and Akkaynak et al. [40], who proposed the sea-thru method for underwater image water removal through the estimation of the backscatter and an optimized framework on RGBD images. Lu et al. [41] and Han et al. [42] reviewed more aspects of underwater optical processing, including underwater image de-scattering, restoration, and quality assessments, with an overview of future trends and challenges in underwater image processing. For the aim of this work, the algorithm developed by Peng and Cosman [9] was chosen: the image blurring and light absorption (IBLA) algorithm is an improved version of the blurriness prior technique, able to estimate more accurate background light and underwater scene depth to further restore the images under various types of complicated scenarios. More implementation details are reported in Section 2.5.

In recent years, deep learning techniques have been successfully applied to computer vision tasks, such as image recognition, segmentation, or object detection. Researchers are more and more involved in applications related to underwater image enhancement; for example, Li et al. [43] proposed a CycleGAN as a color correction method, while Skinner et al. [44] developed a two-stage neural network for image depth estimation and its subsequent color correction. These methods are usually based on physical imaging models, which make them unable to generalize over various conditions; if this model is not considered, the resulting architectures may have complex structures and harder training processes, while if it is simplified, no significant effects are obtained. For this reason, Chen et al. [7] proposed an improvement of the existing DL-based method, which combines a backscatter and a direct-transmission estimation module in a CNN; the resulting output is fed to a reconstruction module together with the input image to finally obtain the enhanced underwater image. Details on its functioning are reported in Section 3.2.

Following these works and considerations, we present a workflow of image acquisition and processing from underwater video footage at depths greater than 10 m and the related photogrammetric approach for the 3D mapping of the study area. Our results have been used to evaluate the quality of the mapping in areas with Posidonia meadows, providing measurable and classifiable data in 3D (especially in the case of matte). The use of the IBLA algorithm and the deep learning CNN-based method further allowed for improved classification accuracy at a glance, with the operators being able to better distinguish the coverage areas and estimate their percentage, as shown by the results discussed in Section 4. This approach demonstrates the potential of underwater photogrammetry in generating highly accurate biocenotic maps for any further research projects aimed at habitat preservation and protection.

## 2. Materials and Methods

### 2.1. Study Area

The survey was conducted in the marine area of the “Spiaggia Nera” in Maratea (PZ), Italy, in the month of May 2022, which is the period of maximum flowering of the *Posidonia oceanica*.

Two separated regions of interest were delineated through the analysis of the bathymetrical and geomorphological data: (i) a sandy area characterized by a relevant presence of matte with a considerable height, at a depth of about 6–8 m; and (ii) a surface with a predominant presence of biocoenosis, characterized by a sandy bottom interspersed with rocks and matte, at a depth of about 14 m.

### 2.2. Instrumentation

The ROV used for the investigations was the MINI ROV-UD 6-100 from the IN.TEC S.r.l. company (the H.Q. are located in Rome, Italy), specifically designed to quickly perform a variety of basic underwater tasks: observation, inspection, exploration, and monitoring. Moreover, the following sensors are mounted on the ROV: (i) an integrated system with a 3-DOF gyroscope, a 3-DOF accelerometer, and a 3-DOF magnetometer; (ii) an internal barometer; (iii) a pressure/depth and temperature sensor; (iv) a current and voltage sensor; and (v) a system for leak detection.

For the scope of this survey, Tritech’s MicronNav positioning system (designed for small systems such as the VideoRay ROV) was used. This system includes (i) a submarine MicronNav transponder/responder, (ii) a surface USBL transducer with integrated magnetic compass and pitch/roll sensors, (iii) a surface MicronNav 100 interface hub, and operating software in control of a host PC/laptop. It has a tracking range of 500 m and 150 m in the horizontal and vertical directions, respectively. Such system is built and distributed by Tritech International Limited, which is in Aberdeen, UK.

Two GoPro Hero cameras, a 3+ and an 8+, were used for the video acquisition. The GoPro 8+ was mounted on the ROV with the camera axis tilted by about 10° with respect to the horizon line; the GoPro 3+ was instead used by the diver in a face-down camera setting [34]. The cameras are built and distributed by GoPro’s Inc, its Head Quarter is located 3025 Clearview Way, San Mateo, CA-USA.

Both of them were set in autofocus mode to acquire a video stream at a resolution of 1920 × 1080 pixels with a framerate of 59.94 fps.

### 2.3. Survey Methodology

Indirect survey techniques were used for the preliminary survey of the extension of the meadows. Firstly, a multi-beam and a side-scan sonar were used for the bathymetric data acquisition, which was processed to obtain the digital terrain model (DTM) (Figure 1a) and the side-scan photomosaic (Figure 1b). These maps were accurately studied to characterize the seabed and identify areas of interest to carefully plan the joint diver/ROV mission. It has to be noticed that the cell size of the MBES-DTM is 0.5 m, with a potential overall accuracy that can be estimated to be around 0.1 m. On the other hand, the mean ground sample distance (GSD) of the photogrammetric process was less than 0.01 m, which makes it not quite comparable with the DTM cell size.

In this study, a series of transects were detected and further navigated to acquire the video. To allow for the correct scaling of the 3D model, the diver positioned two cylindrical target bars 1 m long in two strategic points of the area of interest; these targets were painted in red and yellow for easy identification when working in shallow waters. During the processing phase of the photogrammetric model, one of the two bars was used as a constraint, while the second was used as a control to verify the deformations of the model. To enhance the reliability of the scaling in the three dimensions, we tried to place a vertical scale bar in the area; unfortunately, the bottom morphology did not allow its stable positioning throughout the entire acquisition. This issue, together with the oscillations induced on the scale bar by the movements of the ROV and of the diver, led us to place it on the bottom to avoid the possibility of worsening the result.

In this work, analyses were conducted on (i) transect SN2, taken with the Go Pro Hero 8 and acquired from the ROV at 14 m depth following the height of the mat, and (ii) video of transect ST7, acquired by the diver with a Go Pro Hero 3 while maintaining a distance from the seafloor of about 3 m. To assure the acquisition of an appropriate image overlap, a speed of about one knot was kept constant during the navigation [34]. The acquisition and positioning software (PDS2000) integrated with the USBL allowed to track the ROV position during its navigation.

### 2.4. Photogrammetric Workflow

To obtain a three-dimensional model of an area of interest, several images acquired from different points of view were processed using a generic photogrammetric workflow, as shown in Figure 2.

The first stage of this workflow consists of the setup of both the environment to be surveyed and the ROV (or robot chosen for the scope). The latter can be armed during the second phase to start the image or video acquisition of the area, generally in high-definition (HD) video mode when using action cameras, e.g., GoPro. Once acquired, the videos undergo a preprocessing step for the extraction of single frames, which constitute the first batch of data, i.e., the original data. Then, one or more additional datasets are created from the enhanced images obtained using different algorithms for the correction of underwater-related image disturbances. At this point, each dataset is imported into the software chosen for the elaboration, where the images are processed to extract the tie-points and to estimate the camera pose (image orientation); during this stage, the camera calibration is further adjusted [45]. Once all the cameras have been oriented, a dense point cloud can be computed using algorithms of image matching that involve epipolar geometry [46]. From the dense point cloud, a triangular mesh is obtained, and reliable area measurements of the region of interest can finally be made.

For this specific survey, a scuba diver placed two calibrated bars on the seafloor for the environment setting, while the ROV was equipped with a pre-calibrated action camera of the GoPro series set on a linear FOV. The pinhole camera model and the classical Brown’s distortion model [47] were used to depict the lens distortions: Figure 3 reports (a) the radial distortion obtained by the on-the-job camera calibration procedure and (b) the residuals on the images.

At this point, the video was processed to extract the frames for the original frame dataset, on which two different algorithms were applied: the IBLA and the CNN-based ones (the latter based on a deep learning technique); further details are reported in the next section. Both algorithms allowed to produce enhanced images with a sensible reduction of the blue component, which usually affects images acquired in underwater environments. The obtained dataset was imported into the Agisoft Metashape Software version 1.7.1 for the extraction of the tie-points and the camera pose estimation, with the consequent adjustment of the camera calibration with the so-called bundle adjustment. At the end of this stage, a dense point cloud was computed for each of the three image datasets, and the related triangular mesh was obtained. Metric considerations and area characterization were finally made for the best two models, as shown in Section 3.

### 2.5. Algorithms for Image Enhancements

Figure 4 shows the interaction between light, transmission medium, camera, and scene. The camera receives three types of light energy in line of sight (LOS): the direct transmission light energy reflected from the scene captured (direct transmission), the light from the scene that is scattered by small particles but still reaches the camera (forward scattering), and the light coming from atmospheric light and reflected by the suspended particles (background scattering) [48]. In the real-world underwater scene, the use of artificial light sources tends to aggravate the adverse effect of background scattering. The particles suspended underwater generated unwanted noise and aggravated the visibility of dimming images.

The imaging process of underwater images can be represented as in Equation (1): if we define J as the underwater scene, t as the residual energy ratio after J was captured by the camera, B as the homogenous background light, then the scene captured by the camera I can be represented as in Equation (1)), which is considered the simplified underwater imaging model (IFM).
(1)$$I^{c}\;\left(x\right)={\;J}^{c}\;\left(x\right)\;t^{c}\left(x\right)+B^{c}\left(1\;–{\;t}^{c}\;\left(x\right)\right)$$

Here, x represents one particular point (I, j) of the scene image, c is one of the red, green, and blue (RGB) image channels, and $J^{c}\;\left(x\right)t^{c}\left(x\right)$ and $B^{c}\left(1\;–{\;t}^{c}\;\left(x\right)\right)$ represent the direct transmission and background scattering components, respectively.

The first restoration method, selected in the context of this paper, is based on both image blurriness and light absorption, where more accurate background light (BL) and depth estimations are provided. In particular, Peng, Y. T. et al. [9] proposed a workflow in which the BL is first estimated from the blurry regions in an underwater image, and on its basis, the depth map and then the transmission maps (TMs) are obtained to restore the scene radiance.

In particular, the blurriness estimation process was previously proposed by the same authors and includes three steps: An initial blurriness map is computed starting from the original image, to which a spatial Gaussian filter is applied. They apply a max filter to calculate the rough blurriness map, which is further refined by filling the holes caused by flat regions in the objects through morphological reconstruction; a soft matting or a guided filtering finally allows for the smoothing and generation of the final blurriness map. The BL determines both the color tone and the restored scene radiance of the underwater image, with a brighter scene if a dim BL is used and an opposite result if a brighter BL is chosen. Moreover, a small value in one of the color channels of the estimated BL will lead to a substantial increase in that color in the restored image. In general, the value for the estimated BL of an underwater image is chosen from faraway scene points with high intensity. In contrast, we estimate the BL based on image blurriness and variance. They propose a BL candidate selection method that picks three BL candidates from the top 0.1% blurry pixels in the input image and both the lowest variance and the largest blurriness regions; these latter (which may or may not be the same) are decided using quadtree decomposition, which iteratively divides the input image into four equal-sized blocks according to the variance or blurriness. The blurriness of a region in the input image is obtained by averaging the corresponding region in the blurriness map. They then pick the BL for each color channel separately according to the input image, which is calculated by a weighted combination of the darkest and brightest BL candidates. At this point, the depth estimation is obtained through the combination of three separate subprocesses, sigmoidally combined on the basis of the lighting and image conditions in which each of them better performs.

As it relates to the TM, it is calculated according to the Beer–Lambert law of light attenuation, which uses the depth from the camera to scene points and the related distance between the closest scene point and the camera. This produces more satisfactory restored results with better contrast and saturated color, and it also deals with the artificial lighting. Please refer to the original paper [9] for a more detailed explanation of the IBLA method.

Similarly, the idea at the basis of the DL approach proposed by [7] for underwater image enhancement and restoring stands in the use of convolutional neural networks to fit the multiple components of the image formation model: it is in fact composed of three different sections that estimate (i) the ambient light, which is further used to obtain the (ii) direct transmission map.

More in detail, the backscatter estimation module includes two groups of 3 × 3 convolution kernels, one global mean pooling layer, and two groups of 1 × 1 convolution kernels.

The parametric rectified linear unit (PReLU) is chosen as the activation function for the convolution operation since it avoids the zero-gradient effect in specific cases in which the weights cannot be updated, which can instead occur when the normal rectified linear unit (ReLU) is used.

The direct transmission estimation module concatenates the backscatter estimation with the input image and then applies three groups of 3 × 3 dilated convolutional kernels, which can enlarge the receptive field by filling extra zeros in the kernel without adding more parameters, and one group of 3 × 3 normal convolutional kernels. Different dilation rates are selected for consecutive dilated convolutions in a hybrid dilated convolution, which allows to avoid the gridding effect.

The training of this CNN sees the input image and the corresponding reference image fed to the network, on the basis of which a mean square error is calculated on the pixel value. This is further used as the loss function between the enhanced image and the target one, while the Adam optimizer is used to optimize neural network parameters.

## 3. Results

### 3.1. Initial Image Processing

The videos were fragmented to extrapolate the frames, which we then processed using two commercial software programs widely employed for photogrammetric applications: PIX4D (v. 4.6.4) and Agisoft Metashape (v. 1.7.1). This choice was made to compare the reliability and repeatability of the results provided by the two software in terms of the number of correctly oriented images, the calculation time for the extraction of the dense point cloud, and the number of points in the same cloud. Both enabled the alignment of the frames using the classical structure of motion algorithms. As previously mentioned, the accurate identification on several frames of the two 1 m bars also allowed to correctly scale the photogrammetric model. As a result, the PIX4D elaboration of the ST7 transect (280 frames) generated a dense cloud of about 2.2 million points, with a mean resolution on the seafloor of 2 cm/pixel. The SN2 transect (688 frames), instead, produced a dense cloud of almost 4 million points with a mean ground resolution of 4 cm/pixel. Pix4D was used for the production of 3D models because of the peculiarity offered by this software of being able to build the dense point cloud by setting the multiplicity of images on which all points of the dense cloud appear.

The photogrammetric model was scaled on one of the two scalebars; the second one was used as a check distance, providing a residual of 10 mm. This result allows to estimate a local accuracy of 1%, which satisfies the ISPRA requirements at first glance but should be viewed critically due to the lack of vertical scaling. From the dense point cloud, a mesh was determined in order to obtain a clear distinction between Posidonia, opaque rock, and sand (Figure 5). The final point cloud was then imported into the CloudCompare open-source software (v. 2.12.3), which allowed to classify the sea bottom and accurately dimension the Posidonia and the matte. Some preliminary results are reported here. We measured the sea bottom extension on the 3D model obtained from the ST7 transect, which resulted in a length of 25 m and a width of about 5 m. We also distinguished the presence of Posidonia, matte, sand, and rock: there is a predominance of rocks alternated with small sand banks, from which the Posidonia extends up to about 1.4 m. Figure 6 shows the 3D model of the SN2 transect with a well-defined vertical profile of the matte of about 2 m. The foot of the matte does not show signs of undermining, and there are areas with tufts of *Posidonia oceanica* scattered on the wall—a sign of the vitality of the meadow.

To correctly characterize the nature of the coverage of the area of interest, the two point clouds were manually classified in CloudCompare, where the following layers were identified: “matte”, “Posidonia”, “sand”, “rock” and “unclassified”. Areas of uncertain characterization on the 3D model obtained from the dense point cloud were placed in the latter layer.

Figure 7 shows the different classified areas for both the SN2 and the ST7 transects; especially for the former, there is a consistent area that was not possible to characterize.

### 3.2. Analysis of Image Enhancement Results

To identify the best enhancement algorithm between the IBLA and the CNN-based one, a detailed analysis of the results obtained on the frame processing belonging to the SN2 transect was made using Agisoft Metashape. This choice was made because Metashape can process more images in a much shorter time than Pix4D.

The 3384 original frames were elaborated with a fixed camera calibration by preselecting the “medium accuracy” and the “generic selection” methods; a total of 3182 frames were aligned. From the block of aligned images, a dense point cloud (with “medium” quality and “average” filter) was extracted. The results obtained by using the same setting and employing the IBLA and CNN-based image enhancement algorithms are reported in Table 2.

As can be seen from Table 3, the IBLA-processed images were aligned for a total number that is slightly higher than the original and the CNN-derived frames. However, the latter method allowed for the creation of a more consistent, sparse cloud. Moreover, by comparing the three images (Figure 8), it is evident that the DL-based enhancements perform highly with respect to the others. For this reason, we have chosen CNN as the restoring algorithm for the creation of the final 3D model to be used for the sea-bottom coverage. The next section reports the detailed results obtained by the image processing performed in Agisoft Metashape and used in this work for the characterization phase.

### 3.3. Deep Learning Images Enhancements and Characterization Results

The application of the CNN-DL method to the images effectively removed the bluish and greenish effects from the video images, giving a more contrasted chromatic definition with respect to the various investigated surfaces (see Figure 8, Figure 9 and Figure 10).

The obtained dense point clouds were then imported into the Cloud Compare (v. 2.13.alpha) software to be recharacterized (Figure 11 and Figure 12). Cloud Compare is part of an open-source project; it is one of the most widely used 3D point cloud and mesh processing software. As a first step, the cloud is accurately cleaned, with the consequent creation of the mesh to be used for the definition of the total surface of the considered area. Subsequently, different classification polygons are defined for each biocenosis, in particular Posidonia, matte, and sediment. After creating the mesh, the surface area in m² is calculated for each identified polygon.

By analyzing the PIX4D elaboration of the SN2 and ST7 transects, a better definition of the data is obtained, as reported in Table 3.

The same basic morphologies were identified by redefining the surfaces on the CNN-based 3D models. The classification of the seabed was simpler thanks to the clear distinction of the morphology contours, especially in the case of the Posidonia and matte coverage (see Figure 13 and Figure 14). In fact, the epiphytic species present on them are also clearly visible on the matte, defined by a more accentuated color.

Table 4 and Table 5 report the results obtained for the transects SN2 and ST7 before and after the application of the CNN-based algorithm, respectively. The total area that has been possible to characterize in the second case results in being wider, especially in the case of the SN2 transect (with an increase of more than 100 m^2^ for the SN2 and of 70 m^2^ for the ST7); moreover, the non-classified area undergoes a sensible decrease for the SN2 transect, with a total zeroing for the ST7. These results confirm the great contribution given by the CNN-based enhancement algorithm, which allowed for an easier and more accurate analysis of the seabed and its characterization.

Figure 15 shows the characterization of the SN2 transect after the enhancement: as confirmed by the numerical evaluation, the Posidonia coverage results are greater than those measured in the first analysis, with a lower presence of matte and the same percentage of sand.

Figure 16 reports instead the biocoenosis analysis made on the ST7 transect 3D model obtained through the CNN-DL algorithm. Also in this case, a better definition of the colors allowed to redefine the coverage percentage, with an increase in the Posidonia over the matte and a clearer definition of the presence of rocks, which was not clearly distinguishable in the original 3D model.

This is due to the difficult interpretation of the original data, which led to the neglecting of some uncertain areas; the enhanced data allowed instead to perform a more accurate classification with the consequent reintegration of the previously deleted areas, which further resulted in an overall increase of the classifiable surface (especially for the SN2 transect).

This enhanced data will further allow to obtain information on the age of the Posidonia, confirming the potential of the employed techniques (ROV navigation and GoPro cameras for the visual data acquisition) for the accurate characterization of the sea bottom and its coverage, especially if a CNN-DL-based algorithm is applied to ease the interpretability of the 3D model.

## 4. Conclusions

In this work, we present a photogrammetric application for the underwater monitoring, mapping, and reconstruction of the *Posidonia oceanica* meadow. The use of multi-beam and side-scan sonar allowed us to investigate the structural and morphological characteristics of the sea bottom and plan the survey to be performed by the ROV and diver.

The elaboration of the frames acquired by the GoPro Hero cameras, used in these tests produced point clouds that were scaled using the two sized targets strategically placed in the surveying area. We obtained an accurate reconstruction of the 3D model, on which we measured the dimensions of the Posidonia and the matte, dead matte, or other substrate present at the bottom. Moreover, the application of mesh to the dense point cloud allowed us to estimate the percentage of Posidonia coverage in the investigated area.

The use of the IBLA algorithm and the deep neural network allowed to obtain remarkable enhancements in the 3D model analysis, in particular as they relate to the colors and the characterized area. This can be seen from the images at first and is further confirmed by the elaboration of the point clouds. The reconstruction obtained through the DL approach improved the overall consideration possibilities and was chosen for the final sea-bottom analysis. Furthermore, the different layers were easily identified, providing a more accurate characterization of the surface and of the Posidonia extension. Future works could be based on a supervised technique to enhance the actual characterization methodology, but more data will be needed to properly train the NN, so this work aims at providing a starting point towards the development of this automated approach.

The dimensional study of the Posidonia mattes will allow to obtain quantifiable data for long-term monitoring; moreover, future works could aim to identify suitable areas for Posidonia transplantation. Another aspect to consider is the possibility of creating a correlation between the size of the matte (vertical growth, definition of the percentage of dead or living matte) and the potential of the matte to act as a biogeochemical sink (blue carbon). In this context, future works will be focused on the reliability of the vertical scaling through the use of a reference cubic target in order to assure the accurate sizing of the matte height.

## Figures and Tables

**Figure 1 jimaging-09-00113-f001:**
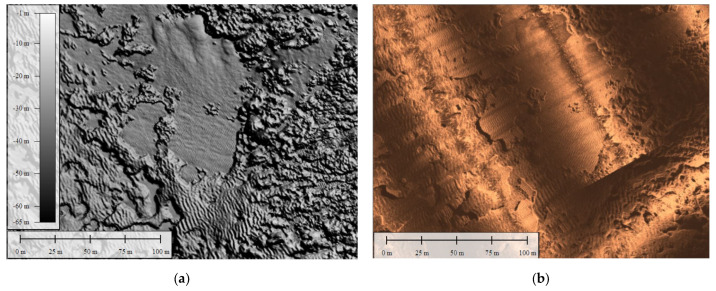
*Posidonia oceanica* meadows, Maratea area: characterization of the seabed, digital terrain model (DTM) (**a**), and side-scan sonar photo-mosaic processing (**b**).

**Figure 2 jimaging-09-00113-f002:**
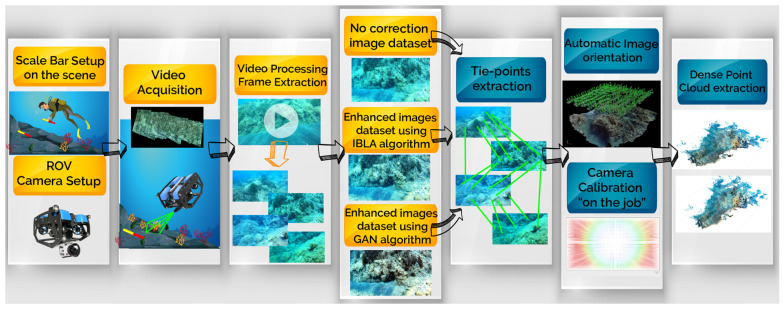
Photogrammetric workflow, the diagram illustrates the process adopted in this work.

**Figure 3 jimaging-09-00113-f003:**
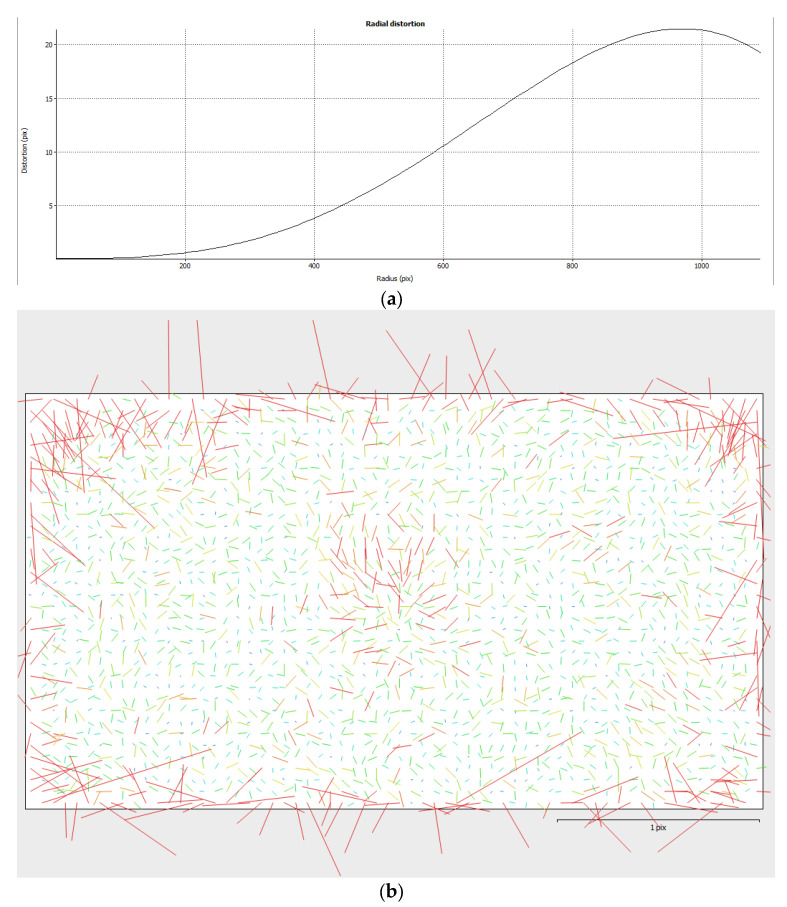
Radial distortion plot (**a**) and residual (**b**) on the image sensor obtained at the end of the on-job camera calibration process.

**Figure 4 jimaging-09-00113-f004:**
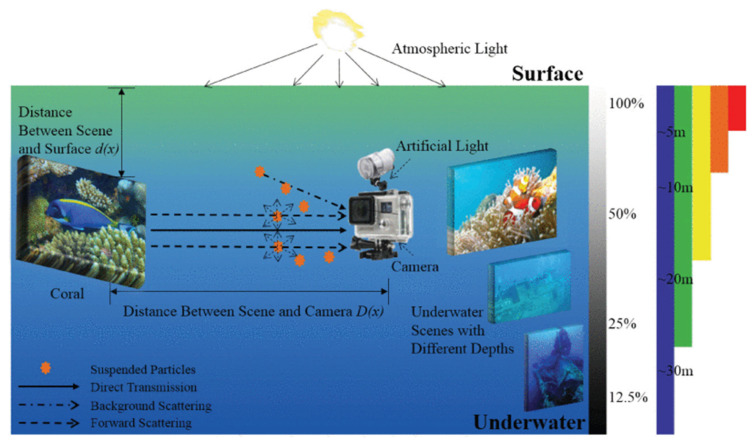
Underwater optical imaging process and the selective attenuation of light (adapted from [49]).

**Figure 5 jimaging-09-00113-f005:**
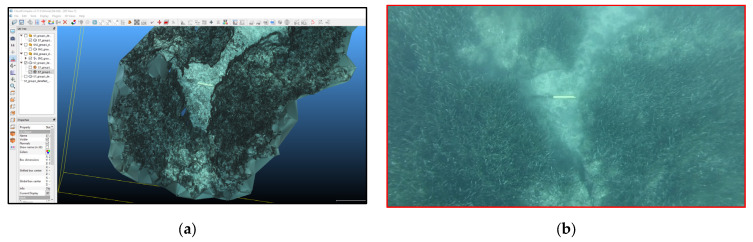
Section of the points cloud (**a**) and corresponding frame (**b**).

**Figure 6 jimaging-09-00113-f006:**
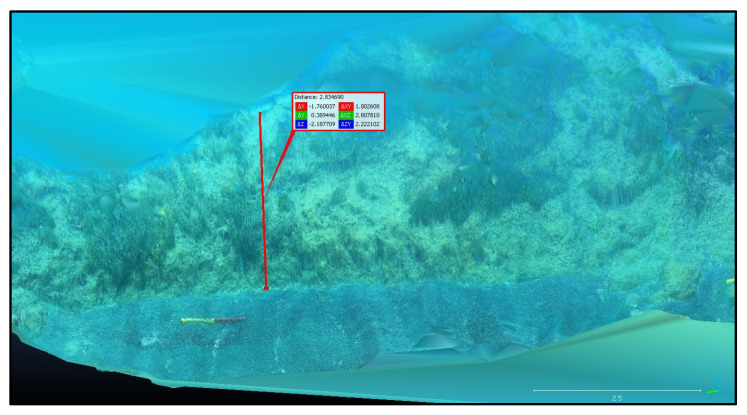
Three-dimensional model of the SN2 transect showing the measure of the vertical profile of the matte.

**Figure 7 jimaging-09-00113-f007:**
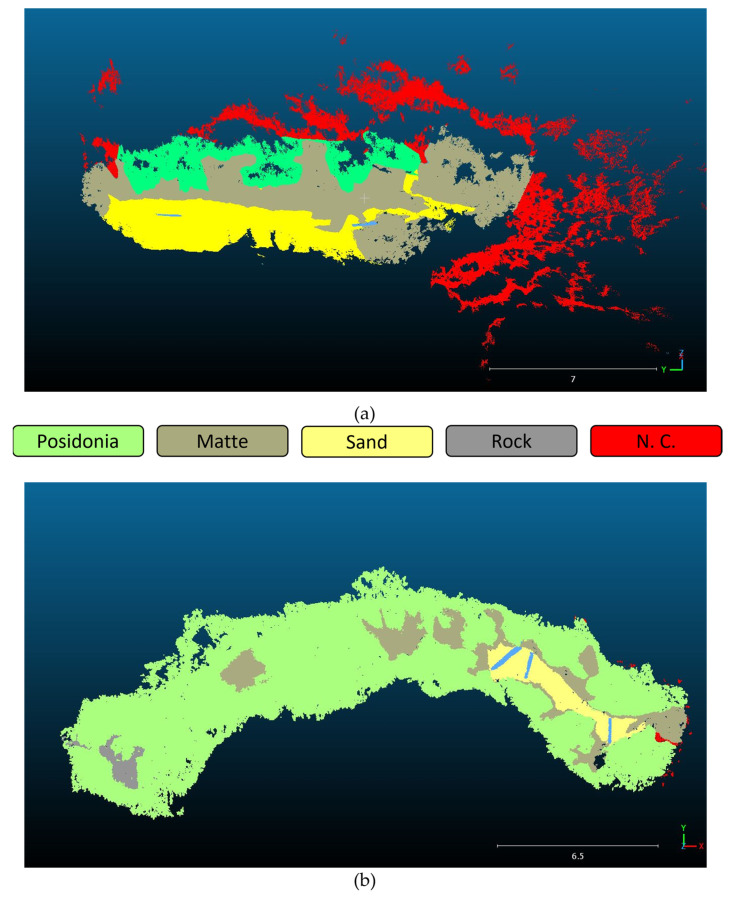
Characterization of the SN2 (**a**) and ST7 (**b**) transects on the basis of the identified layers. The targets are depicted in red.

**Figure 8 jimaging-09-00113-f008:**
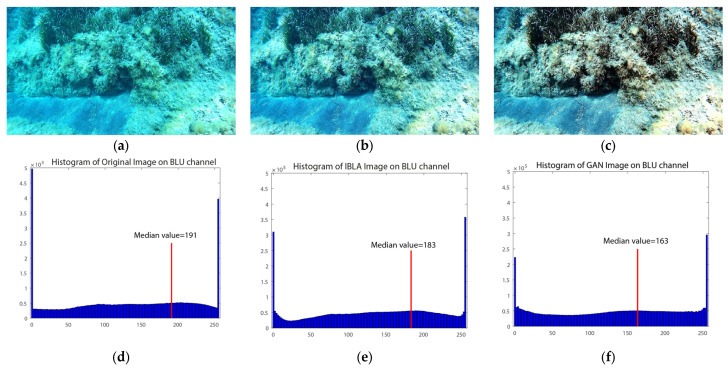
A comparison of the two different enhancement algorithms: (**a**) the original image; (**b**) the frame processed with IBLA; and (**c**) the frame processed using CNN. The corresponding histograms reported the variation of the blue channel with respect to the original image (**d**), IBLA (**e**), and CNN (**f**). For each histogram, the median value was reported and illustrated with a vertical red line.

**Figure 9 jimaging-09-00113-f009:**
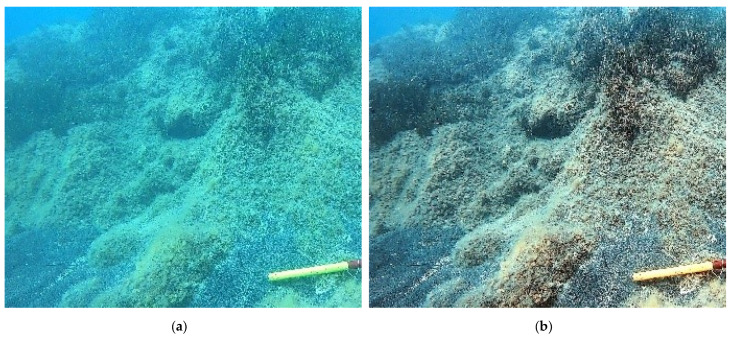
SN2 transect—comparison between the frame GH1049_2605_1347_04_frame00345_512 before (**a**) and after (**b**) the enhancement.

**Figure 10 jimaging-09-00113-f010:**
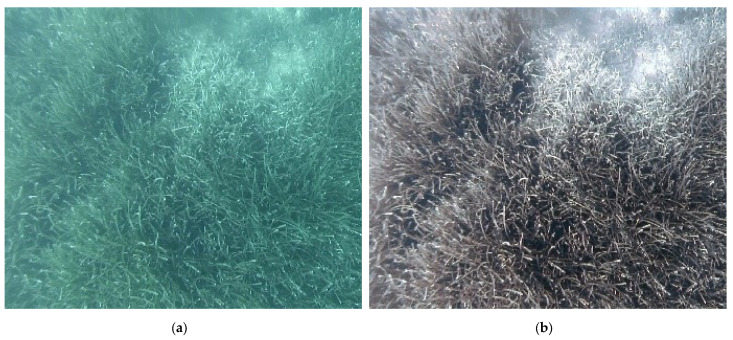
ST7 transect—comparison between the same frame GOPR0404_frame00227_512 before (**a**) and after (**b**) the enhancement.

**Figure 11 jimaging-09-00113-f011:**
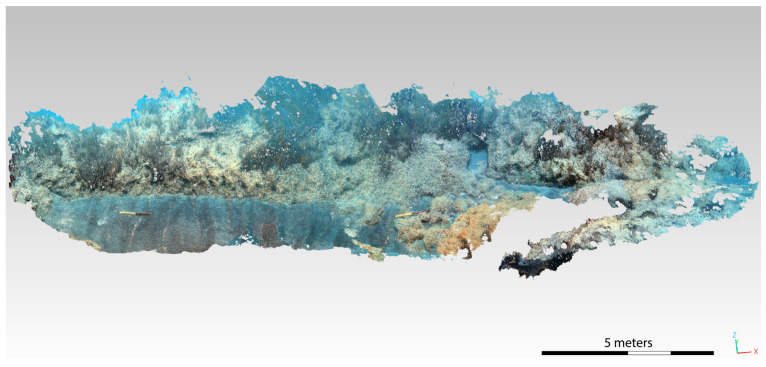
Three-dimensional model of the SN2 transect after the application of the CNN-DL algorithm.

**Figure 12 jimaging-09-00113-f012:**
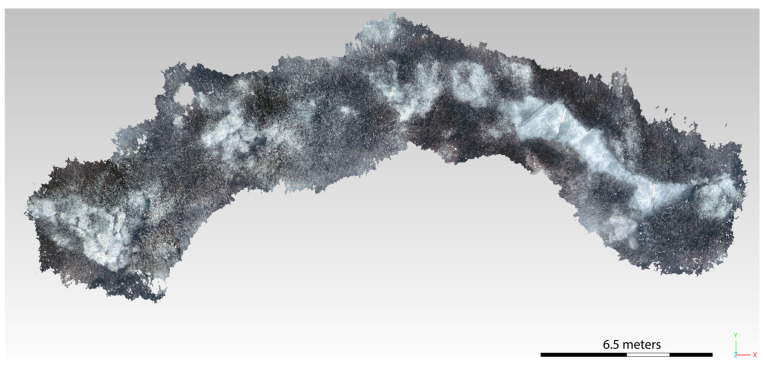
Three-dimensional model of the ST7 transect after the application of the CNN-DL algorithm.

**Figure 13 jimaging-09-00113-f013:**
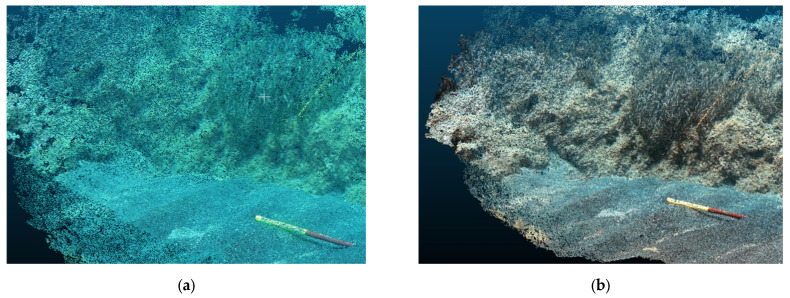
SN2 transect—comparison between a detail of the 3D model before (**a**) and after (**b**) the enhancement.

**Figure 14 jimaging-09-00113-f014:**
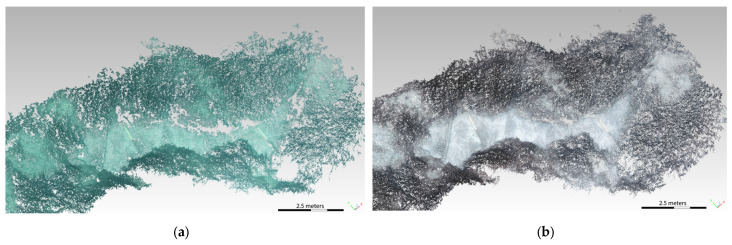
ST7 transect—comparison between a detail of the 3D model before (**a**) and after (**b**) the enhancement.

**Figure 15 jimaging-09-00113-f015:**
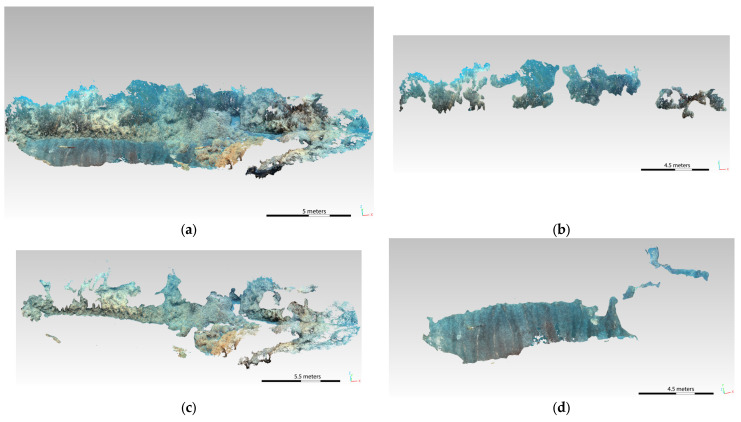
SN2 transect—identification of the habitat from the 3D model obtained from the enhanced frames processing (**a**): sea bottom coverage of Posidonia (**b**), matte (**c**), and sand (**d**).

**Figure 16 jimaging-09-00113-f016:**
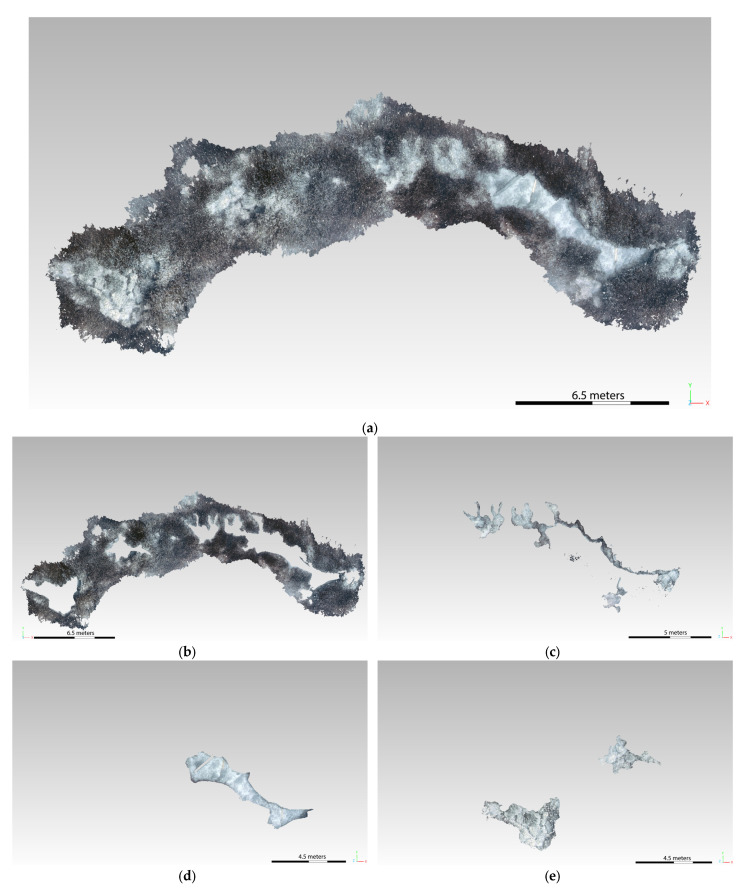
ST7 transect—identification of the habitat from the 3D model obtained from the enhanced frame processing (**a**): sea bottom coverage of Posidonia (**b**), matte (**c**), sand (**d**), and rocks (**e**).

**Table 2 jimaging-09-00113-t002:** Comparison of the elaboration of the original and enhanced frames.

	Original Frames	IBLA Frames	CNN Frames
N° total images	3384	3384	3384
N° aligned frames	3182	3230	3216
N° sparse cloud points	335,852	335,131	356,055
N° dense cloud points	5,696,743	5,543,270	5,678,596

**Table 3 jimaging-09-00113-t003:** Comparison of the results obtained for the two transects before and after the application of the enhancement (PIX4D software).

	N. of Aligned Frames	N. of Dense Point Clouds
SN2	688	39,71,913
SN2_CNN	688	4,444,126
ST7	280	2,204,910
ST7_CNN	280	2,661,009

**Table 4 jimaging-09-00113-t004:** Coverage area estimation of the SN2 and SN7 transects before the image enhancement.

Transect	Total Area	*Posidonia oceanica*		Matte		Sand		Rock		Non-Classified	
	[m^2^]	[m^2^]	(%)	[m^2^]	(%)	[m^2^]	(%)	[m^2^]	(%)	[m^2^]	(%)
SN2	307.15	55.422	18	110.320	36	50.217	16	/	/	90	29
ST7	415.77	351.903	85	45.469	11	12.782	3	5.416	1	1.23	0.3

**Table 5 jimaging-09-00113-t005:** Coverage area estimation of the SN2 and SN7 transects after the image enhancement.

Transect	Total Area	*Posidonia oceanica*		Matte		Sand		Rock		Non-Classified	
	[m^2^]	[m^2^]	(%)	[m^2^]	(%)	[m^2^]	(%)	[m^2^]	(%)	[m2]	(%)
SN2	425.008	95.406	22	181.904	43	50.217	12	/	/	99.462	23
ST7	485.859	416.12	86	32.09	7	16.69	3	23	4	0	0

## Data Availability

The data presented in this study are available on request from the corresponding author.

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
