# Peer review of "An Enhanced Photogrammetric Approach for the Underwater Surveying of the Posidonia Meadow Structure in the Spiaggia Nera Area of Maratea"

_2313-433X, 2023, doi:10.3390/jimaging9060113_

Round 1

Reviewer 1 Report

The article describes the optimization of image-based 3D reconstruction of underwater scenes using structure from motion by a-prior correction of the underwater image colorization. The classification of the resulting 3D models is evaluated with respect to the underlying coloration of the single points in the point cloud. 

All in all, the topic is really interesting, the text is well written and supported with meaningful illustrations. The literature review is extensive and complete.

Unfortunately, methodological consideration of photogrammetric 3D object point reconstruction and classification were completely ignored. The reader is told in passing that Agisoft Metashape (and Pix4D?) were used but a description of the underlying method and the chosen parameters (camera parameterization, camera model, image acquisition configuration, etc.) is missing. There are also methodological weaknesses: The scaling of a 3D model can never be done without doubt with a single scale bar serving as  single source for ground control. At least one degree of freedom remains. The second scale bar, used for verification, lies in exactely the same plane, which is why any errors in, e.g., depth direction go unnoticed. The applied methods should be thoroughly tested and documented accordingly.

Furthermore, the structure of the paper needs revision. In the section "Experiments", some decisive methodological steps are described (rather casually), which the reader would rather have expected in the method section. 

Until the end of the paper, it was not clear to me what the purpose of the study was supposed to be. In the end, different 3D models were created, different algorithms for image pre-processing were tested and the model of the Meadow was finally classified to derive information about it. However, the suggested title of the Manuscript is "An enhanced photogrammetric approach for the underwater ....". So I expected a detailed description of one specfic workflow rather than a study of different workflows. Is the focus on modeling the Posidonia meadow or on examining different workflows to determine point clouds and subsequently surface models of specific underwater landscape forms? I recommend stating specific research questions that will be answered in the paper.

I put some detailed comments in the PDF that may help the authors to revise the Manuscript.

English language sounds good to me. However, I´m not a native speaker and cannot judge the quality.

Author Response

Dear Reviewer,

Thank you for your comments to our work. We revised the manuscript according to your suggestions and clarified some points. Please find a list of our modifications and motivations below, following your comments on the pdf file.

  • Line 19. Two major effects of the underwater environment on the images have been added.
  • Line 35-37. Reference added.
  • Line 75. We clarified that we are referring to the accuracy required by the ISPRA guidelines.
  • Line 125-127. We specified which positioning technique allows for the use of the GNSS underwater (Acoustic Systems)
  • Line 166. The reference you suggested has been added.
  • Line 218. A section containing the photogrammetric approach and the workflow we followed has been added in the materials and methods chapter (see section 2.4).
  • Figure 1. The figure has been shifted in section 2.3.
  • Line 252. Thank you for your comment, we specified the resolution of the DTM and the overall accuracy estimation, highlighting that by virtue of its value this data could not serve as an independent reference.
  • Line 257. This is perfectly accurate, and we are conscious of the consequence, unfortunately, we did not have the possibility of placing a vertical scale bar on the sea bottom. Thanks to your comment we have explained this issue in text. As you highlighted, we cannot estimate the overall accuracy of the entire 3D model, but we have estimated the scale accuracy just locally. Of course, a reference data can provide an accuracy estimation of the 3D model, but the size of DTM cell is too big for carrying out a reliable comparison.
  • Line 259. We used red and yellow scale bars because these colors are easily identifiable in a bluish environment; moreover, we are working in shallow waters, so the absorption effect is supposed to be at its lowest.
  • Line 286. Typo corrected.
  • Line 324. Both the IBLA and the CNN-based color correction methods have been explained in the section since they have been both used on the images, as introduced in the first lines of the manuscript.
  • Line 349. The choice of using two different software came from the willing of comparing the reliability and repeatability of the results provided by the two software in terms of number of correctly oriented images, calculation time for the extraction of the dense point cloud and number of points in the same cloud.
  • Line 360. Thank you for your comment, we modified the text in order to clarify that the estimated accuracy is valid locally.
  • Line 375-378. The characterization of the point cloud has been manually made to meet the requirements of the Italian authority (ISPRA). Further work is being made on unsupervised classification techniques.
  • Line 380. Yes, we are used to work on slices but in this case we had the need of correctly picking the points: the possibility to rotate the mesh was particularly helpful, thus we measured the distance directly from the points cloud.
  • Line 388: Thank you for your observation. We modified the title of the section specifying that it will describe the results of the image enhancement analysis.
  • Line 406: Thank you for your comment. We have analysed and compared the results obtained by different images datasets using Agisoft Metashape, sorry for the mistake.
  • Line 412. This title is clearly ambiguous and has been changed.
  • Line 421. We had manually classified the point clouds before and after the image enhancement. The software version has been correctly moved.
  • High resolution images with clearer scales have been provided. We also modified the background following your suggestion. Thank you.

Thank you for your valuable suggestions.
The authors

Reviewer 2 Report

Your paper describes results of the application of point cloud segmentation based on underwater image photogrametry (followed by improvement techniques).

You paper reads well and is nicely documented (references). However it is supprisingly suffereing from quantitative results.

As an average user of Machine learning technique, I would suggest to use metrics such as Confusion matrix, ROC ...

Figures 6,7,8; 9,10;11,12; 13-14 notably, should integrate more of these metrics (or others that I do not know of and might be more appropriate) in order to further document the improvement of the enhancements made by your method.

With the results described in a more robust way your paper will largely gain in scientific soundness and significance.

Author Response

Dear Reviewer,

Thank you for your comments to our work.

We sincerely understand your proposal on the use of evaluation metrics as a mean to improve the robustness of our work and its scientific soundness. However, the characterization of the sea bottom that we presented in this work is completely manually made by the user to meet the requirements of the Italian Authority (ISPRA), so no computed evaluations could have been made. Moreover, being this the first digitalization output of the Spiaggia Nera area, no pre-existing data is available for comparison and metrics calculation; in fact, our approach not only aims at improving and supporting the actual methodologies recognized in the field but wants to place the basis for further developments towards the use of Deep and/or Machine Learning for supervised or unsupervised classification. We are in fact already working on the assessment of the best automized classification technique for this kind of data, which will be hopefully presented soon.

Our scope is to highlight the valuable contribution of restoring algorithms to the enhancements of the 3D point cloud and to the consequent comprehensive categorization of the seabed, which becomes then easier and more efficient than that obtained by traditional methods, which is an important breakthrough in the field.

Thank you again for your valuable suggestions.
The authors

Reviewer 3 Report

I would like to extend my congratulations to the authors for their work.

I find the methodology presented in this paper for monitoring and mapping the Posidonia oceanica meadows to be quite interesting. The use of underwater photogrammetry, combined with the application of two different algorithms, including a Deep Learning approach, is a novel approach to addressing the challenge of environmental factors affecting underwater images. The 3D point cloud obtained using the restored images allows for a more comprehensive categorization of the seabed than traditional methods, which is an important breakthrough.

It is worth noting that the correct application of the methodology presented in this paper is crucial to achieving accurate results. The researchers have taken great care to account for the various environmental factors that can impact underwater images and have utilized advanced algorithms to minimize their effect on the final 3D point cloud.

Overall, I believe this work has the potential to make a valuable contribution to the field of marine ecology and conservation. The ability to rapidly and reliably characterize the Posidonia coverage is critical for assessing the health of marine ecosystems and preserving coastal morphology, and this methodology offers a promising solution to this challenge.

Author Response

Dear Reviewer,

We would like to sincerely thank you for your appreciation of our work.

The authors.

Round 2

Reviewer 1 Report

I would like to thank the authors for the significant revision of their manuscript which has greatly increased the quality of the document. I still have a few comments (Line Number: comment):

L: 135/136: Reference for GNSS use missing (for Ultra-Short and Long Base-Line). The reader might be not familiar with these techniques and would like to read some more information about it.

L: 266-268: Why are the parameters of the DTM are not comparable with the photogrammetric DTM? Because of the expected resolution or single point accuracy? Please provide the reason why you think, that the bathymetric DTM is not usable as a reference or for evaluating the accuracy of the generated photogrammetric DTM(s).

L: 276-282: Too bad that the vertical scalebar could not be installed. In the future, the use of floats and attachment to the ground could be considered. (Since we are only looking at scale, even a slight change in position due to currents would be negligible). I strongly expect that the absolute result would improve many times. 

L: 307: The right R at the GoPro designation can be omitted from my perspective (was also not attached earlier in the text).

L: 320: Was the Brown-Model sufficient to estimate the lens distortion of the wide-angled, almost fish-eye, GoPro camera? From experiments in the past (with GoPro 4 Hero Black) we know that modeling lens distortion with the GoPro is tricky (and have switched to using other camera models, e.g. equidistant models, to describe lens distortion; to de-sample the images and only continue working with the corrected images in e.g. Metashape). As a reader, I would be highly interested to see, for example, a distortion plot (before and after undistortion).

L: 422-424: I don´t get the point of this sentence. What do you mean by "by setting the multiplicity of images on which the generic point appears."? What is "the generic point" in this context. 

L: 429: Please mention that the specified accuracy meets the claims at first glance, but should be viewed critically due to the lack of vertical scaling. 

L: 445: You wrote that the classes have been determined manually, which is ok. However, I was wondering if Machine-Learning approaches (e.g. Random Forest) might be an option to classify underwater point clouds? Could this be an idea to be considered in future work (e.g. when acquiring point clouds of larger underwater areas)?

L: 579: Please be careful to talk about GoPro cameras in general here. New GoPros use different sensors, different lenses, have different setting options for image capture, etc. Please only refer your conclusions to the actual systems used.

L 600: When talking about future work and the estimation of vertical growth, vertical scales are indispensable. As already suggested, please address the methodological improvement suggestions in the future work section (e.g. 3D scaling). 

General comments:

Please check the quality of your images (unfortunately the pictures often look very pixelated).
Fig. 11 is only partly visible (not sure if this is the fault of the authors)

Author Response

The authors thank the reviewer for the significant work done.

Please see the attachment for the answers to 2nd round.
